# Synthetic genetic oscillators demonstrate the functional importance of phenotypic variation in pneumococcal-host interactions

Anne-Stéphanie Rueff[1], Renske van Raaphorst [1,2], Surya D. Aggarwal [3], Javier Santos-Moreno [1,6], Géraldine Laloux [2], Yolanda Schaerli [1], Jeffrey N. Weiser[3] & Jan-Willem Veening [1,4,5] ✉

Phenotypic variation is the phenomenon in which clonal cells display different traits even under identical environmental conditions. This plasticity is thought to be important for processes including bacterial virulence, but direct evidence for its relevance is often lacking. For instance, variation in capsule production in the human pathogen *Streptococcus pneumoniae* has been linked to different clinical outcomes, but the exact relationship between variation and pathogenesis is not well understood due to complex natural regulation. In this study, we use synthetic oscillatory gene regulatory networks (GRNs) based on CRISPR interference (CRISPRi) together with live cell imaging and cell tracking within microfluidics devices to mimic and test the biological function of bacterial phenotypic variation. We provide a universally applicable approach for engineering intricate GRNs using only two components: dCas9 and extended sgRNAs (ext-sgRNAs). Our findings demonstrate that variation in capsule production is beneficial for pneumococcal fitness in traits associated with pathogenesis providing conclusive evidence for this longstanding question.

Phenotypic variation, in which clonal cells display strikingly different phenotypes even under identical environmental conditions, plays a role in many bacterial bet-hedging and division-of-labor strategies[1–10]. Single cell technologies have revealed heterogeneity in gene expression within clonal bacterial populations, but direct evidence for the functional significance of this variation is limited. One of the best understood examples linking phenotypic variation to benefits during virulence is bistable expression of the Salmonella pathogenicity island 1 (SPI-1)[11–14]. Noise in gene expression creates subpopulations of genetically identical bacteria that differ in their level of virulence[15]. The slower-growing, SPI-1-expressing subpopulation is less susceptible to antibiotics and invades the host, benefiting the non-virulent subpopulation[11,13,16].

The exopolysaccharide capsule is the main virulence factor of *S. pneumoniae*[17], also known as the pneumococcus. This bacterium is responsible for over 500,000 deaths per year[18]. Interestingly, capsule production displays reversible ON/OFF switching in many clinical strains, leading to opaque (O) and transparent (T) colony variants[19–21]. The O variant (encapsulated) is more virulent in systemic infection, while the T variant (unencapsulated) is a better colonizer[20,22]. Capsule levels depend on phase variation of the DNA methylomes through inversions of three homologous methyltransferases[23–25]. Different variants of the methyltransferase methylate different DNA recognition sequences causing global changes in gene expression, including capsule expression[23,24]. In addition to phase variation, capsule production and capsule shedding are highly regulated and several transcription

[1]Department of Fundamental Microbiology, Faculty of Biology and Medicine, University of Lausanne, Biophore Building, CH-1015 Lausanne, Switzerland. [2]de Duve Institute, UCLouvain, 75 Avenue Hippocrate, 1200 Brussels, Belgium. [3]Department of Microbiology, New York University School of Medicine, New York, NY, USA. [4]Department of Pediatrics, University of California San Diego, La Jolla, CA, USA. [5]Skaggs School of Pharmacy and Pharmaceutical Sciences, University of California San Diego, La Jolla, CA, USA. [6]Present address: Pompeu Fabra University, Barcelona, Spain. ✉e-mail: Jan-Willem.Veening@unil.ch

factors directly bind and control transcription of the capsule operon (*cps*)[26–34]. Promoter swap studies suggest that this controlled and dynamic regulation of *cps* fine-tunes capsule levels during infection[30,35]. Indeed, *S. pneumoniae* reduces its capsule during adherence to epithelial cells while increases capsule production during systemic infection to prevent phagocytotic killing[36,37]. Together, it is generally assumed a suite of environmental and internal signals lead to temporal heterogeneity in capsule expression. However, direct evidence for the hypothesis that phenotypic variation in capsule production is advantageous for the pneumococcal lifestyle is missing.

One approach to understanding natural systems is the use of synthetic biology[38–42]. To test the hypothesis that phenotypic variation in capsule production is important for the pneumococcal lifestyle, we used synthetic GRNs based on CRISPR interference (CRISPRi) to uncouple capsule production in *S. pneumoniae* serotype 2 strain D39V from its native regulation. CRISPRi uses a dead Cas9 enzyme (dCas9) and an sgRNA to silence genes by blocking transcription[43,44]. CRISPRi circuits, in contrast to transcription factor-based synthetic GRNs, impose a low burden on host cells due to the use of short orthogonal sgRNA sequences[45]. CRISPRi was recently used to build synthetic oscillators in *E. coli*[46–48].

Here, we engineered a three-node CRISPRi-based ring oscillator stably encoded on the pneumococcal genome that displays robust oscillations. By rewiring pneumococcal capsule production to the synthetic GRNs, we demonstrate that temporal variation in capsule production is beneficial for fitness during murine colonization, answering the long-standing question that capsule heterogeneity is important for pathogenesis. Our results demonstrate the potential of engineered GRNs to mimic and test the biological functions of natural phenotypic variable systems in vivo.

## Results

### Extended sgRNAs to direct dCas9

The first aim of this study was to replace the natural complex heterogenous regulation of pneumococcal capsule production by dynamic synthetic GRNs in which we can control the levels of phenotypic variation. A well-known dynamic GRN is the so called 'repressilator' in which three orthogonal transcription factors repress each other to generate periodic expression of green fluorescent protein[49]. Recent CRISPRi versions of such a three-node ring circuit in the synthetic biology workhorse *E. coli* showed robust oscillations with periods of multiple hours, much slower than the cell division cycle[46–48]. Since it takes several generations to produce and lose the capsule in *S. pneumoniae*[50], the timescales involved in the repressilator circuit would be well suited to dynamically switch between capsule ON and OFF phenotypes. We have previously successfully used CRISPRi to control gene expression in *S. pneumoniae* and design multistable and dynamic circuits in *E. coli*[46,51,52]. For the *E. coli* circuit that was named the 'CRISPRlator', the Csy4 RNase of *Pseudomonas aeruginosa*, also known as Cas6f, was used to cut sgRNAs from the multigene mRNA[46]. This allowed to have a modular setup with putting binding sites of well-characterized sgRNAs downstream of a promoter. However, our attempts at expressing Csy4 in *S. pneumoniae* failed, probably due to toxicity. Therefore, we considered a different strategy that would allow us the same flexibility without using Csy4 (Fig. 1a): the use of extended sgRNAs (ext-sgRNAs) cassettes that encode a dCas9 binding site upstream of the sgRNA spacer sequence. Thus, DNA sequences encoding ext-sgRNAs can be repressed via CRISPRi but can also be transcribed as ext-sgRNA and bind to dCas9 to repress another ext-sgRNA cassette. Ext-sgRNAs containing a 24 bps extension sequence at their 5' end were cloned with a spacer sequence targeting the firefly luciferase gene, *luc* (Fig. S1a). The extended sequence contains the transcription start site (+1) from a strong constitutive promoter (P3)[53] as well as a protospacer adjacent motif (PAM) followed by a unique orthogonal 20 bps binding site sequence (BS) not present in the

*S. pneumoniae* genome[54]. Out of the six tested ext-sgRNAs, four were still fully functional in repressing *luc* transcription upon dCas9 induction (Fig. S1b). This provided proof of principle that certain ext-sgRNAs retain the capacity to direct dCas9 to targets encoded within the spacer region.

### Construction of a three-node ring genetic oscillator in *S. pneumoniae*

Facilitated by the ability of *S. pneumoniae* to take up exogenous DNA via competence, we constructed the 'CRISPRlator' strain (strain VL3757) in which ext-sgRNA1 represses transcription of ext-sgRNA2, which in its turn represses transcription of ext-sgRNA3 that represses ext-sgRNA1. In addition, we included three spectrally distinct fluorescent proteins each containing a specific BS recognized by one of the ext-sgRNAs in their 5'UTRs (Fig. S1c and Fig. 2a). All parts were present as single copy, stably integrated on the pneumococcal chromosome via double crossover at neutral loci (see Methods). Ext-sgRNA1 represses expression of mScarlet-I and ext-sgRNA2, ext-sgRNA2 represses mTurquoise2 and ext-sgRNA3, and ext-sgRNA3 represses mNeonGreen and ext-sgRNA1 (Fig. 1b). This GRN is expected to produce its output in the following order: mScarlet-I (red), mNeonGreen (green) and mTurquoise2 (blue) (Fig. 1b). *S. pneumoniae* CRISPRlator cells were grown in C + Y medium at 34 °C within a microfluidic device with the mother machine design[55], but customized for the pneumococcus (see Methods and Fig. 2b–c) and observed by time-lapse fluorescence microscopy (Fig. 1c, Movie S1). Analysis of CRISPRlator cells within such devices exhibited periodic expression of fluorescent proteins in the expected red-green-blue order (Fig. 2d). To assess whether the observed gene expression patterns correspond to oscillatory behavior, we analyzed and quantified fluorescence and cell cycle parameters of thousands of single CRISPRlator cells over $42 \pm 21$ (median ± standard deviation) generations to calculate the so-called autocorrelation function that measures the relationship between the current gene expression and its past values. If the autocorrelation function shows a peak at regular intervals, this indicates that the signal is oscillating with a periodic behavior. As shown in Fig. 1d, the CRISPRlator demonstrates a regular autocorrelation function, suggestive of robust oscillations with an average period of $590 \pm 210$ (median ± standard deviation) min corresponding to $11 \pm 1.41$ cell divisions. By tracking individual cells over multiple cell divisions, until they were washed out of the microfluidic chamber, we could generate lineage trees superimposed by fluorescence signals highlighting dynamic gene expression of the three fluorescent proteins (Fig. 2e). By following a single cell, and arbitrarily selecting one of its daughter cells after division for over a period of 50 h, clear cell cycle independent oscillation in the order red->green->blue was evident (Fig. 1e). Together, these results show that we successfully constructed a genome-encoded CRISPRi-based oscillator using ext-sgRNAs in *S. pneumoniae* that shows periodic oscillations similar to previously reported plasmid-based repressilator circuits in *E. coli*[46–48].

### CRISPRi-controlled heterogeneity in pneumococcal capsule production

Next, we aimed at rewiring capsule production under the control of the synthetic oscillator. The pneumococcal *cps* operon is located between conserved genes *dexB* and *aliA*, and all genes required for capsule synthesis are driven by a primary promoter and a weaker, secondary internal promoter (Fig. 3a)[56]. To control the *cps* operon with the CRISPRlator, we replaced the native primary *cps* promoter for a constitutive promoter and inserted BS6-*mScarlet-I*, which is targeted by ext-sgRNA1, resulting in the 'CAPSUlator' (strain VL4315) (Fig. 3a). Thus, when cells express mScarlet-I, the *cps* operon is transcribed and the capsule produced. Conversely, when ext-sgRNA1 levels are high, capsule production is switched off (Fig. 3b). Next,

three control strains were constructed: (i) a strain lacking ext-sgRNA3 in which cells always repress *cps* and mScarlet-I and constitutively express mTurquoise2 and mNeonGreen (strain VL4322 'CAPSUlator-OFF'), (ii) a strain that oscillates but completely lacks the *cps* genes (strain VL4321 'CAPSUlator-Δ*cps*'), and (iii) a strain lacking

the correct BS upstream of *cps* and thus constitutively expressing mScarlet-I and the capsule (strain VL4324 'CAPSUlator-ON') but still oscillating for mTurquoise and mNeonGreen (Fig. 3b). Immunofluorescence showed that the CAPSUlator displays phenotypic variation in capsule production that correlated with mScarlet-I

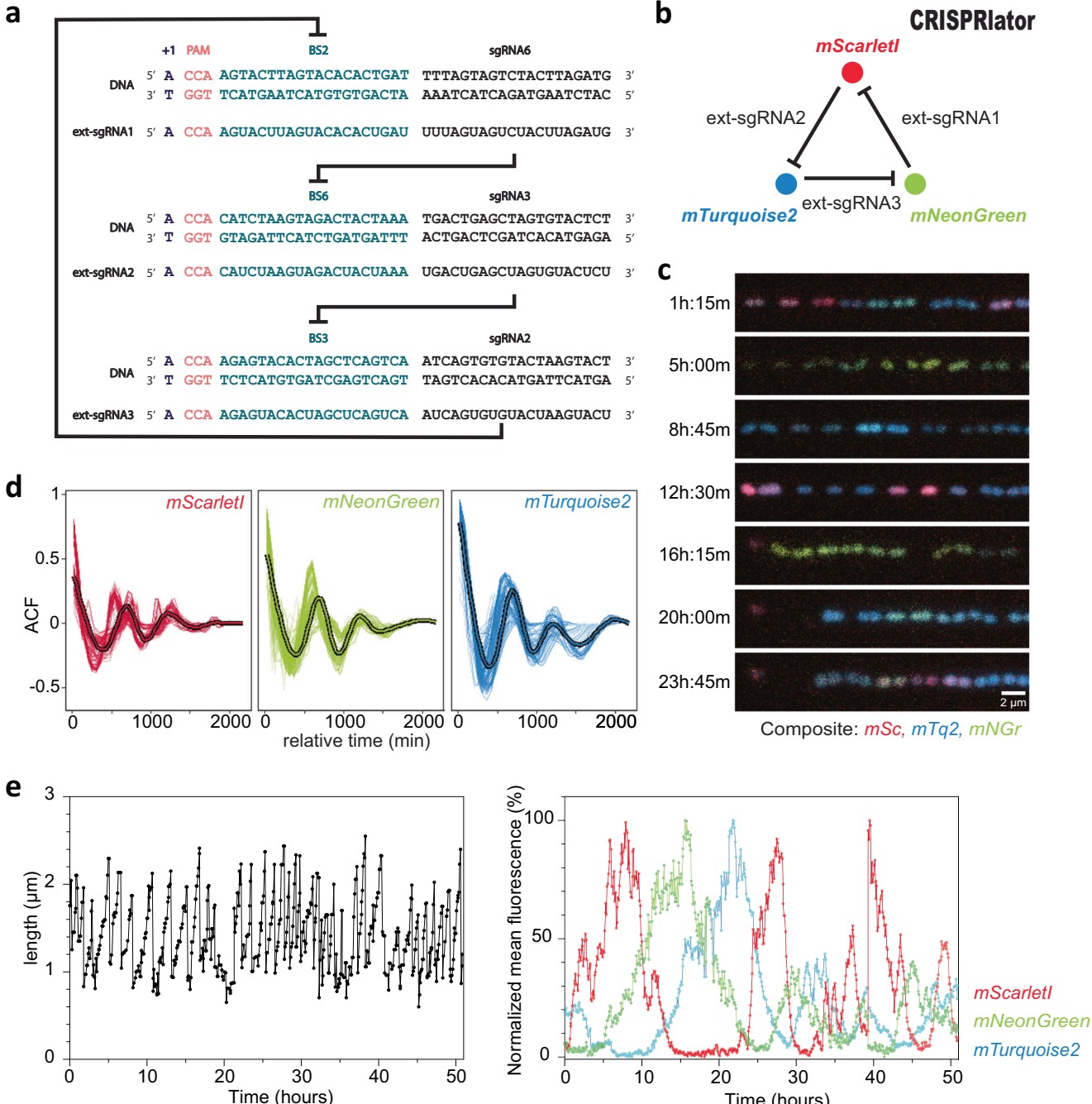

**Fig. 1 | Design and characterization of a three-node ring oscillator 'CRISPRlator' using extended sgRNAs. a** Schematic overview of the use of ext-sgRNAs to construct a three-node ring oscillator. Ext-sgRNA1 has a spacer sequence (sgRNA6) that is complementary to binding site 6 (BS6) of the non-template strand of the DNA sequence encoding ext-sgRNA2. Ext-sgRNA2 has a spacer (sgRNA3) targeting BS3 present on the DNA sequence of ext-sgRNA3. Ext-sgRNA3 has a spacer (sgRNA2) targeting BS2 present on the DNA sequence of ext-sgRNA1. For clarity, only the variable 5′ region of ext-sgRNAs sequences is shown (the dCas9-binding handle and transcription terminator are not shown). **b** Graphical representation of the three-node ring oscillator called CRISPRlator. Expression of ext-sgRNA1 represses transcription of the mScarlet-I reporter (red); ext-sgRNA2 represses mTurquoise2

(blue) and ext-sgRNA3 represses mNeonGreen (green). As the ext-sgRNAs also repress each other's transcription (**a**), this GRN is expected to periodically express the three reporters in the order red->green->blue. **c** Snapshots with 3h45min intervals of a typical microfluidics experiment of the CRISPRlator strain are shown. Scale bar 2 μm. Time in hours and minutes. **d** Autocorrelation function (ACF) of the CRISRlator cells grown in mother machine channels. ACF calculated in one mother machine lane for individual cells (thin lines) and averaged (fat line with black outlines) shows oscillations for all three fluorescent signals. **e** Progression of cell length (left) and progression of normalized fluorescent signals (right) of one individual mother cell in the mother machine over time. Source data for **d** and **e** are provided as a Source Data file.

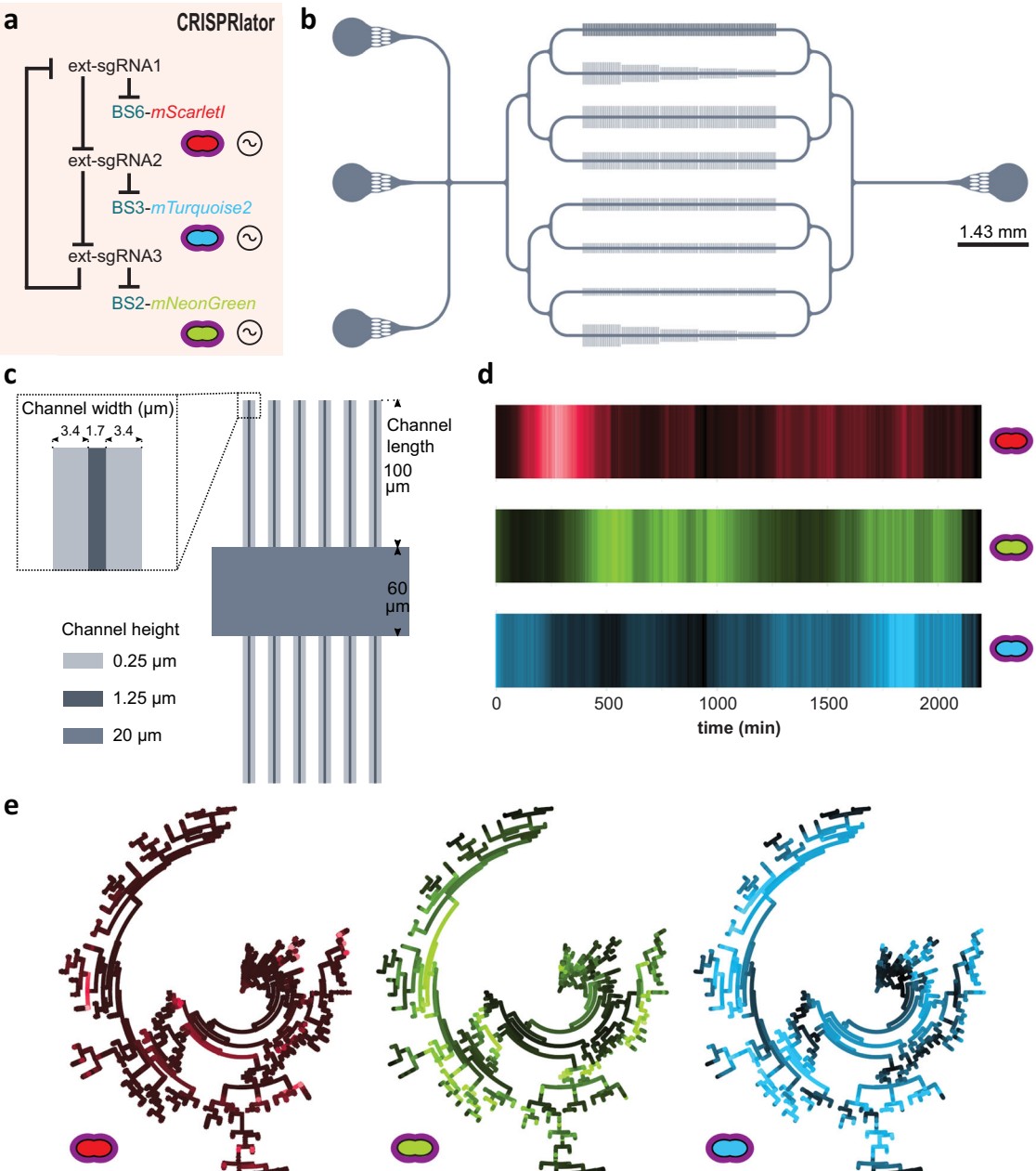

**Fig. 2 | Characterization of the pneumococcal CRISPRlator. a** Expression of ext-sgRNA2 represses transcription of the mTurquoise2 reporter (blue) and ext-sgRNA3; ext-sgRNA3 represses mNeonGreen (green) and ext-sgRNA1, and ext-sgRNA1 represses mScarlet-I (red) and ext-sgRNA2, leading to oscillatory behavior (⊣-symbol). **b** Schematic overview of the here designed and used microfluidics device. The chip, measured from the outer edge of the inputs/output, is 14.3 × 6.8 mm. Bacteria were injected via the top inlet and C + Y media was pumped through the middle inlet at a flow rate of 0.5 ml/h. The bottom left inlet was blocked. The single waste outlet is shown on the right. Scale bar = 1.43 mm. **c** Zoom in on a mother machine lane within the microfluidic device. Mother machine channels are 1.7 μm wide, 1.25 μm high and have 3.4 μm wide side channels of 250 nm height to increase nutrient/waste flow. The main feeding/flow channel is 20 μm high and 60 μm wide. **d** Mean expression of a total lineage of cells over time in one mother machine lane shows red-green-blue oscillation over time. **e** Circular lineage tree of the same population shown in **d**, where color intensity is the mean fluorescence intensity of each single cell in the lineage (left-right: mScarletI, mNeonGreen and mTurquoise2). The ancestor cell is positioned in the middle of each tree.

expression, while CAPSUlator-OFF and CAPSUlator-Δ*cps* lacked capsule (Fig. 3b, c and Fig. S2). As expected, CAPSUlator-ON showed homogenous capsule and mScarlet-I production, but heterogenous mTurquoise2 and mNeonGreen expression (Fig. S2). Note that in vitro growth in rich C + Y medium was similar for all strains except a slightly slower growth of the constitutively expressing capsule CAPSUlator-ON strain, as expected (Fig. S2e).

To examine whether the CAPSUlator showed oscillatory behavior in capsule production, time lapse fluorescence microscopy was performed within microfluidics mother machines. As shown in Movie S2 and Fig. 3d, red, capsule producing cells, occupy the channels in single row while blue, unencapsulated bacteria can also grow in double row within the microfluidics devices. This confirms the immunofluorescence experiments and demonstrate that red cells are encapsulated and thus occupy more space than blue, unencapsulated bacteria. Together, these experiments show that we could employ a CRISPRi-based three-node ring GRN to drive oscillatory behavior in pneumococcal capsule production.

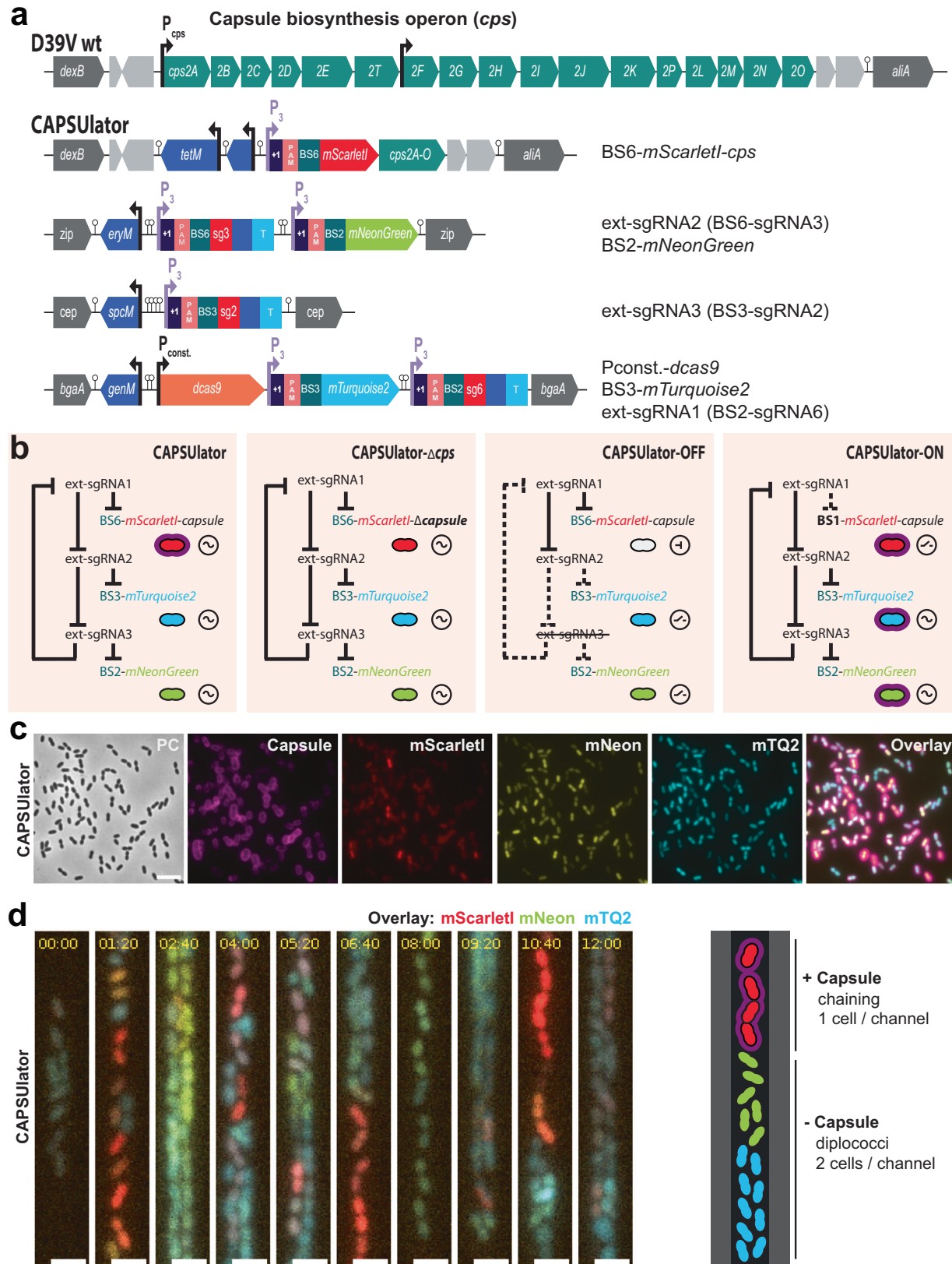

## Phenotypic variation in capsule production benefits pneumococcal survival and virulence

The first step in pneumococcal infection is nasopharyngeal colonization[57]. To test the hypothesis that heterogeneous capsule expression can be beneficial for the pneumococcal lifestyle, we first looked at the capacity of the CAPSUlator strains to adhere to abiotic surfaces that mimics biofilm formation, a crucial process for colonization[58]. Bacteria were grown in liquid C + Y medium till OD 0.4 and subsequently diluted 100x, transferred in 96-well plates and grown at 34 °C with 5% $CO_2$. After 6 h, the supernatant was removed, and adhering bacteria were quantified by crystal violet staining (see Methods). While wild-type pneumococci poorly adhere to the

**Fig. 3 | CAPSUlator design and characterization. a** Top: capsule operon and genetic context in wild type D39V. Bottom: CAPSUlator construct placed in four genetic loci on the chromosome. **b** Gene regulatory circuits of the CAPSUlator and control strains. The ⌐ symbol indicates that the three-node oscillatory network is intact. The ⊣ symbol signifies repression and the _/- symbol indicates the ring-node network is interrupted. **c** The CAPSUlator shows heterogeneous production of capsule and the three fluorescent proteins. Capsule production correlates with the production of mScarletI (see also Fig. S2). A micrograph from a representative

experiment is shown (the experiment was performed at least 3 times). Scale bar 5 μm. **d** Left: snap-shots of a microfluidic time-lapse microscopy movie of the CAPSUlator with 1h20m time intervals. Scale bar 2 μm. Cells producing mScarlet-I produce capsule: the cells occupy more space, thereby fitting only one cell in the width of the channel, while cells mainly producing mNeonGreen and mTurquoise2 can fit two cells in the width of a channel and are not growing as a chain. Right: schematic representation.

polystyrene plates, capsule mutants including CAPSUlator-OFF and CAPSUlator-Δcps efficiently adhere (Fig. 4a). The CAPSUlator strain expressing capsule heterogeneously adhered slightly better than wild type ($P = 0.03$), while bacteria that homogeneously expressed the capsule in the CAPSUlator-ON strain were poor biofilm-formers (Fig. 4a).

Next, we assessed the ability of the strains containing the synthetic GRNs to adhere to human nasopharyngeal epithelial cells. Detroit-562 cells were grown to confluence and exponential growing pneumococci were added at an MOI of 5 (see Methods). After 1 h, free floating and adhering bacteria were plated and CFUs were counted the next day. A strain-specific adherence ratio was calculated by dividing the number of adherent bacteria by non-adherent bacteria. As shown in Fig. 4b, the CAPSUlator and CAPSUlator-ON strains had similar adherence ratios compared to wild type, whereas cps mutants and the CAPSUlator-OFF strain adhered significantly better.

The capsule was also shown to be important for survival during starvation as pneumococci can use their capsular polysaccharide as a carbon source[59]. To test how well cells with synthetic control of the capsule cope during starvation, the CAPSUlator strains were grown in C + Y medium to mid-exponential phase. Subsequently, the cells were washed and resuspended in phosphate-buffered saline (PBS) at 25°C, and bacterial viability was recorded over time. As expected, bacteria showed a progressive loss of viability and after 24 h approximately 0.1% of wild type and CAPSUlator cells were still able to form colonies (Fig. 4c, Fig. S2f). The control strains without capsule, CAPSUlator-OFF and CAPSUlator-Δcps showed even further reduced viability and less than 0.01% of cells were alive after 24 h (Fig. 4c). On the other hand, approximately 1% of CAPSUlator-ON bacteria were still alive after 24 h (Fig. 4c).

While in vitro assays favor specific phenotypes (e.g., adherence favors unencapsulated bacteria while immune evasion favors encapsulated bacteria), it is unclear what the optimal strategy is in a relevant in vivo animal model where both adherence and immune evasion are required. To examine if phenotypic variation in capsule expression is indeed functional in vivo, we used an infant mouse model of colonization. In this model, bacteria require adherence to the nasal epithelium and also immune evasion for effective dissemination and transmission[60]. Four-day-old pups were inoculated intranasally with $10^5$ CFU of pneumococci. After 24 h the pups were sacrificed, and bacterial loads were enumerated in nasal lavages (see Methods). As shown in Fig. 5a, wild type D39V as well as the CRISPRlator strain (with wild type capsule regulation) were present at similar colonization levels, demonstrating that the presence of the synthetic CRISPRi GRN with the fluorescent reporters has no detrimental impact to in vivo pneumococcal fitness. Strikingly, the CAPSUlator significantly outperformed the CAPSUlator-OFF, CAPSUlator-Δcps and CAPSUlator-ON network strains. This unambiguously shows that heterogeneous capsule production provides an advantageous strategy compared to homogenously expressed capsule in vivo. Notably, the CAPSUlator does not colonize better than wild type bacteria, suggesting that natural regulation of the capsule is optimized for the variable environmental conditions present in the mouse model.

## Discussion

Synthetic GRNs enabled us to test the concept that phenotypic variation in otherwise clonal bacteria is beneficial under specific conditions. We show that exploiting orthogonal ext-sgRNAs instead of RNA cleavage factors like Csy4 or standard sgRNAs[46,48] allows for the creation of complex GRNs using only two components: dCas9 and ext-sgRNAs. This eliminates the need for additional factors to be expressed or PAM sequences to be inserted at the target site and should advance the design and construction of complex CRISPRi-based GRNs in the future, regardless of host organism. It is interesting to note that the frequency of oscillations shown by the single copy, genome integrated pneumococcal CRISPRlator and CAPSUlator GRNs are very similar in length compared to CRISPRi-based oscillators constructed on multicopy replicating plasmids in E. coli[46–48]. It is tempting to speculate that dCas9 and sgRNA stability, target binding affinity and DNA replication rates play a role in setting the oscillation frequency. Future research should find out what the key parameters are in driving these CRISPRi-based oscillators so that rational engineering of oscillators of various frequencies and amplitudes would become possible.

It is generally assumed that phenotypic variation provides a selective advantage for bacterial virulence. However, empirical evidence supporting this claim is scarce because the signals involved to trigger heterogeneity are hard to model in vitro. Here, we employed a synthetic biology approach to engineer the human pathogen S. pneumoniae and directly address the long-standing question whether heterogeneity in capsule production is important for its life cycle. In several assays that mimic traits involved in pneumococcal virulence such as biofilm formation, adherence and starvation, we observed that different strategies are beneficial. Whereas the absence of capsule improves bacterial binding to abiotic and biotic surfaces, the presence of capsule increases starvation survival (Fig. 5b). However, in an infant mouse colonization model, which requires both adherence and immune evasion, heterogenous capsule production outperformed homogenous capsule strategies (Fig. 5a). Thus, while pneumococci lacking capsule adhere better and form thicker biofilms, they are more readily cleared by the host (Fig. 5b). What the exact dynamics are of the CAPSUlator in vivo still needs to be examined. The presence of the synthetic GRN and all associated fluorescent proteins encoded within the pneumococcal genome did not seem to impose a negative metabolic burden as the CRISPRlator strain showed similar in vitro and in vivo fitness compared to wild type pneumococci (Fig. 5a), reinforcing the idea that CRISPRi has a relatively low burden for the cell[45]. Whole genome sequencing of the CAPSUlator verified the presence of the designed GRN and demonstrated the absence of any suppressor mutations (see Methods). Likely, dcas9 mutants would be rapidly outcompeted by 'wild type' GRNs as such mutants would constitutively express all ext-sgRNAs as well as the fluorescent proteins normally repressed by the dCas9-ext-sgRNA complex. It is interesting to note that CAPSUlator bacteria did not outperform wild type bacteria during murine colonization and might even do slightly worse, although not with a statistically significant difference (Fig. 5a). This implies that we have not captured the ideal expression dynamics of cps and that precise regulation of capsule synthesis in response to the environmental conditions is crucial for optimal colonization. In our synthetic GRNs, capsule is expressed with temporal heterogeneity in which individual

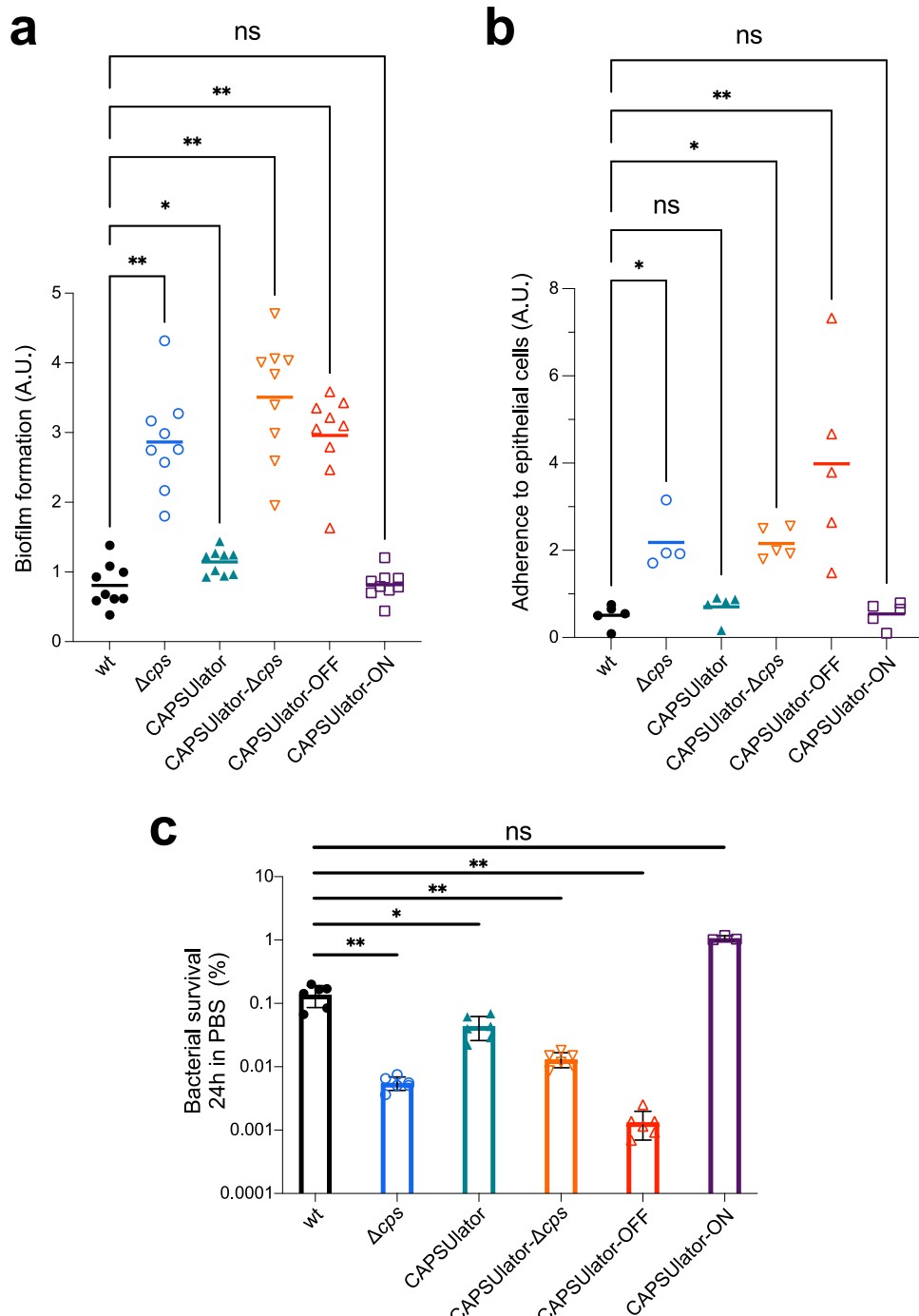

**Fig. 4 | Characterization of synthetic GRNs driving pneumococcal capsule production in traits associated with virulence. a** Pneumococcal biofilm formation was measured by growing strains in microtiter plates at 34 °C. After 6 h, biofilm formation to the wells was quantified using crystal violet staining (see Methods). The amount of biofilm formed by each strain was compared to wild type *S. pneumoniae* D39V using a two-tailed Wilcoxon signed rank test ($n = 9$). **b** The ability of the engineered CAPSUlator strains to adhere to human nasopharyngeal epithelial Detroit-562 cells was tested by infecting a monolayer of cells at an MOI of 5. After 1 h of incubation at 37 °C the non-adherent and adherent bacteria were enumerated by plating (see Methods). The ratio of adherent vs non-adherent bacteria is shown and compared to wild type D39V using a Kruskal−Wallis test ($n > 4$). **c** Bacterial survival during starvation was tested by resuspending exponentially growing cells in 1 x PBS followed by incubation at 25 °C. Viable bacteria were quantified by plating and colony counting (see Methods). After 24 h of starvation, all synthetic GRNs except for the CAPSUlator-ON strain showed significantly reduced survival compared to wild type D39V (two-tailed Mann−Whitney test). *$p < 0.05$, **$p < 0.01$ ($n > 3$). Data are presented as mean values +/− SD. Each symbol represents a biologically independent replicate. Source data are provided as a Source Data file. a.u. arbitrary units.

cells gain and lose capsule over time. It would also be interesting to test and compare alternative GRNs that would create noisy heterogeneity, in which cells randomly express capsule or not, like the case for SPI-I expression in *Salmonella*[13,15]. Such experiments might also provide insights whether division of labor, bet hedging or a combination of such strategies is beneficial for pneumococcal pathogenesis. Overall, the here presented study provides valuable tools for the pneumococcal and synthetic biology research community, such as the

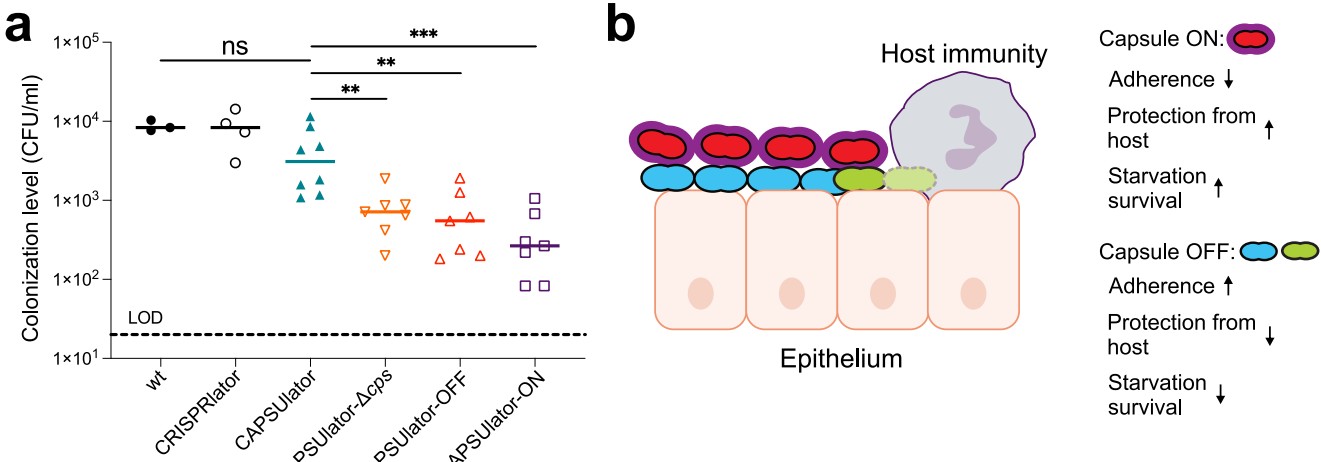

**Fig. 5 | Heterogeneous pneumococcal capsule production is beneficial for in vivo colonization compared to homogenous capsule expression. a** Four-day-old mouse pups were inoculated intranasally with $10^5$ CFU of pneumococci and after 24 h sacrificed and bacterial loads were enumerated in nasal lavages. There was no statistically significant difference between wild type and the CAPSUlator strain, while all other GRNs (CAPSUlator-$\Delta cps$, CAPSUlator-OFF, CAPSUlator-ON) colonized worse than the CAPSUlator (Mann–Whitney test). *$p < 0.05$, **$p < 0.01$, ***$p < 0.001$ ($n > 3$, each symbol represents a biologically independent replicate). Source data are provided as a Source Data file. **b** Conceptual model of the different phenotypes driven by the CAPSUlator GRN and their biological advantages and disadvantages.

implementation of CRISPRi with ext-sgRNAs to construct single copy GRNs, the generation of a mother machine microfluidic device for *S. pneumoniae* (and associated single cell image analysis pipeline), and the demonstration that complex GRNs are functional in vivo in a murine model of colonization. This work may provide a roadmap for analogous systems to study fundamental questions about the roles and evolution of phenotypic plasticity.

## Methods

### Ethics statement
Animal experiments were performed according to the guidelines laid by National Science Foundation Animal Welfare Act (AWA) and the Public Health Service Policy on the Humane Care and Use of Laboratory Animals. NYU's Grossman School of Medicine's Institutional Animal Care and Use Committee (IACUC) oversees the welfare, well-being, proper care, and use of all animals. They have approved the protocols used in this study: IA16-00538.

### Bacterial strains, culture conditions and transformation
All pneumococcal strains in this study are derivatives of *S. pneumoniae* D39V[56]. Strains were grown in liquid semi-defined C + Y medium[61] at 34 °C or 37 °C. Detailed information on strain construction is reported in the Supplementary information. Strains and plasmids are listed in Supplementary Data S1 and oligonucleotides are listed in Supplementary Data S2. The genome of the CAPSUlator strain was sequenced using Illumina technology (Novogene) and is available at SRA (SRR24464804). Transformation of pneumococcal strains was realized after growth in C + Y medium (pH 6.8) at 37 °C until an absorbance (OD$_{595nm}$) of 0.1. Competence was activated by the addition of 100 ng/ml CSP-1 (synthetic competence-stimulating peptide 1) for 12 min at 37 °C. Donor DNA was added to the activated cells and incubated 20 min at 30 °C followed by a dilution 1/10 with C + Y medium and incubated 1 h at 37 °C. Transformants were selected by plating in Columbia blood agar containing the appropriate antibiotic. Final antibiotic concentrations used: 7.5 ug/ml chloramphenicol (chl), 0.5 ug/ml erythromycin (ery), 250 ug/ml kanamycin (kan), 0.5 ug/ml tetracycline (tet), 100 ug/ml spectinomycin (scp), 10 ug/ml trimethoprim (tmp), 40 ug/ml gentamicin (gen).

### Luciferase assays
Pneumococcal strains containing a transcriptional fusion of the firefly luciferase gene (*luc*) were precultured in liquid C + Y (pH6.8) until an absorbance (OD$_{595nm}$) of 0.4, then diluted to OD$_{595nm}$ of 0.004 in C + Y (pH6.8) with luciferin at a final concentration of 0.45 mg/mL and with IPTG (0, 10 or 100 uM). Luciferase assays were performed in 96 flat bottom white polystyrene plate (Corning, 3610) in a microtitle plate reader (Tecan Infinite 200 PRO) at 37 °C without shaking as described before[61]. Optical density (OD$_{595nm}$) and luminescence (relative luminescence units [RLU]) were measured every 10 min in triplicate.

### Microfluidics experiments
Pneumococcal CRISPRlator (VL3757) or CAPSUlator strain (VL4315) was grown in filter sterilized (0.45 µm) liquid C + Y medium (pH6.8) supplemented with 300 U/ml of catalase (Sigma, C1345), 1x Pluronic-F108 (Sigma, 542342) and 100 µg/ml spectinomycin at 37 °C until an OD$_{595nm}$ of 0.04. Growth was continued at 34 °C until OD$_{595nm}$ of 0.14. Prior to injection, the cells were concentrated 1/100 and vortexed to break chains. One of the three input holes of the pneumococcus microfluidic device (Wunderlichips GmbH) was used to inject bacteria with a Hamilton Kel-F Hub needle (Hamilton, HA-90520). A second input hole was used to flow in fresh medium at a rate of 0.5 ml per hour, powered by a syringe pump (World Precision Instruments, AL-1000). The cell input and the unused inlet were closed with a stainless-steel catheter plug (Instech, SP20/12). Cells were grown inside microfluidic chambers for approximately 3 days at 34 °C. Imaging was performed using a Leica DMI8 microscope with a 100x/1.40 oil-immersion objective and a sCMOS camera (Leica-DFC9000GT-VSC08519). Images were taken every 5 min. Phase contrast images were acquired using transmission light (50 ms exposure). Excitation light from a SpectraX (Lumencor) was limited to 50% power output in combination with a 10% neutral density filter through a multipass CFP/YFP/mCherry filter cube (chroma) with an exposure time of 300 ms for each fluorescence channel (mTurquoise2 through the CFP filters, mNeonGreen through the YFP filters and mScarlet-I through the mCherry filters).

### Immunofluorescence
Pneumococcal strains were grown in liquid C + Y medium (pH 6.8) at 37 °C. After two dilutions, bacteria were harvested at OD$_{595nm}$ of 0.1 and incubated with 1/1000 of Pneumococcus type 2 Rabbit antiserum

(SSI Diagnostica, 16745) for 5 min on ice. After incubation, bacteria were washed three times with C + Y medium and then incubated with 1/1000 of goat anti-rabbit IgG antibody, Alexa Fluor 680 (Invitrogen A-1109, A27042) for 5 min on ice. Cells were washed once with C + Y medium and once with ice-cold 1x PBS. Bacteria were concentrated 50x and spotted on an agarose pad (1.2%). Microscopy was performed on a Leica DMI8 with a 100x objective and a SpectraX lightsource at 100% power output. To image the capsule and limit spectral overlap with mScarlet-I, fluorescence was acquired through a filter cube containing an SpX-Q filter set with an excitation of 640 nm and emission wavelength of 720 nm (600 ms exposure time). For mScarlet-I, a SpX-QS filter cube was used with Ex 550 nm and Em 590 nm (800 ms exposure). For mTurquoise2 and mNeonGreen, a Chroma multipass filter cube was used with Ex 440 nm and Em 470 nm (700 ms exposure) and Ex 510 nm and Em 535 nm (600 ms exposure).

## Cell segmentation and tracking

For the microfluidics time lapse movies, cells were segmented based on all 4 channels (CFP, YFP, RFP and phase contrast) using Ilastik[62]: using the pixel classification tool, cells, background and mothermachine were classified manually until the software detected the cells properly. After this, pixel prediction maps of the full movies were generated in bulk mode. Cell masks were generated using the object classification tool. Subsequently, cells were tracked using the Fiji-plugin TrackMate[63,64], using the cell masks as detection input (LAP tracker, frame-frame linking 8 px, gap closing distance 10 px, gap 2 px, track segment splitting distance 8 px). Spots, including the mean intensity per cell mask of each channel, tracks, containing track lengths, and edges, containing information on the relationship between detected masks, were saved as.csv files and imported into R for further analysis. For the snapshots, Morphometrics was used to segment the cells based on phase-contrast[65]. The fluorescence intensity was measured as the mean pixel intensity per cell using the R package BactMAP[66].

## Quantifications and statistical analysis

Data analyses of the mother machine data and the capsule immuno-fluorescence experiments were performed using R. The fluorescence intensity of each channel was normalized between 0-100% to be able to compare oscillations easier. The autocorrelation function (ACF) was calculated over time, per fluorescence channel, per individual cell genealogy, which was defined as the ancestral line of each leaf of the lineage tree. Cell growth was defined as the relative increase in cell length over time. Cell divisions were detected as peaks in cell growth (peak over a span of three time points). Cells that had a generation time that was faster than 3 times the median generation time of 35 min were discarded as misdetections, cells with a generation time longer than 3 times the generation time were discarded as non-growing. The median oscillation times were calculated as the median of the first apparent peak in the ACF functions of each cell genealogy.

Lineage tree visualizations and analysis were done in R using the R packages ggraph and tidygraph (https://github.com/thomasp85/ggraph, https://github.com/thomasp85/tidygraph). Notebooks containing the scripts for the analysis of the microfluidic movies can be found at https://github.com/veeninglab/Capsulator. For the illustration of one individual cell in the mothermachine (Fig. 1e), one single cell was manually tracked using Fiji. The manually tracked mothermachine data was analyzed and plotted using Prism (Graphpad).

## Biofilm assays

Biofilm assays were performed as described[67]. Cells were cultured in C + Y medium (pH 6.8) until OD595nm of 0.4, then diluted 100 times in 200 µl C + Y (pH 7.8) in a 96-well plate (Cytoone CC7672-7596). Plates were incubated for 6 h at 34 °C with 5% CO_2. After incubation, bacterial growth was measured at OD595nm. The supernatant was removed to allow the staining of the remaining biofilm in the bottom and edges of the wells. Biofilms were stained for 15 min at room temperature with 1% crystal violet. Each well was washed twice with distilled water to remove non-adherent cells. Biofilms were solubilized with 200 µl 98% ethanol in each well. Biofilm biomass was quantified by measuring the OD595nm with a microtiter plate reader (TECAN Infinite F200 Pro).

## Starvation survival assays

Pneumococcal strains were grown in liquid C + Y medium (pH 6.8) at 37 °C until an OD_{595nm} of 0.15. Cells were washed once with 1x PBS and then re-suspended in 1x PBS and incubated at 25 °C. Aliquots of each strain were collected over time (0 h, 4 h, 6 h and 24 h incubation) and stored with 16% (vol/vol) glycerol at −80 °C. Viable bacteria were enumerated by diluting the stored aliquots in 1x PBS and plating in triplicate inside Columbia blood agar followed by overnight incubation at 37 °C with 5% CO_2. Colonies were counted manually.

## Adherence assays

Nasopharynx epithelial cells Detroit 562 (Merck 86012804-1VL) were plated in 96 well cell culture plates (Costar 3595). After microscopic observation of the presence of confluent monolayers, cells were rinsed twice with 1x DPBS (Gibco). Pneumococcal strains were grown in liquid C + Y medium (pH 6.8) at 37 °C until an OD595nm of 0.2, then centrifuged and resuspended in RPMI medium 1640 (1x), supplemented with 1% (vol/vol) FCS and 10 mM HEPES (Gibco). Detroit 562 cells and bacteria were co-incubated at a MOI of 5 (i.e., 5 bacteria for every Detroit 562 cell). To optimize the adherence, the plate was centrifuged (at 1000 × g for 5 min) and incubated 1 h at 37 °C with 5% (vol/vol) CO_2. The supernatant was recovered and the cell layer was washed once with 1x PBS to remove non-adherent bacteria. Supernatant and wash were combined and stored with 16% glycerol (vol/vol) at −80 °C as non-adherent fraction (NA). To dislodge the epithelial cells with the adherent bacteria, a solution of trypsin-EDTA was added and incubated for 10 min at 37 °C with 5% (vol/vol) CO_2. The detached cells were collected and washed once with 1x PBS. Detached cells and wash were combined and stored with 16% (vol/vol) glycerol at −80 °C as adherent fraction (A). Both fractions, NA and A, were diluted and plated in 2% (vol/vol) blood Columbia agar. After overnight incubation at 37 °C with 5% CO_2, colonies were counted manually.

## Colonization of infant mice

Infant mice (4 days old) of both sexes were intranasally inoculated with $10^5$ CFU of pneumococcal strains in 3 µl of PBS, without anesthesia. Following intranasal instillation, the pups were returned to their dam. After 24 h, mice were euthanized by CO_2 asphyxiation followed by cardiac puncture. To assess the colonization levels, the trachea was cannulated using a 30-gauge needle and lavaged with 300 µl of sterile PBS collected from the nares. Ten-fold serial dilutions of this retro-tracheal lavage were plated on Tryptic Soy (TS)-catalase plates supplemented with the appropriate antibiotic to enumerate pneumococcal load. Our previous work has shown that there is no difference in the colonization levels between males and females in our infant mouse model. As such, infants of both sexes were used for this work. At least 3 individual pups were used per infection group per experiment.

## Statistics & reproducibility

Non-parametric statistical tests were used to assess phenotypic differences between different GRNs. Sample sizes were chosen to allow for realistic experimental handling as well as having enough statistical power and reduce the use of animals. No data were excluded from the analyses. The experiments were not randomized. The Investigators were not blinded to allocation during experiments and outcome assessment.

## Reporting summary

Further information on research design is available in the Nature Portfolio Reporting Summary linked to this article.

## Data availability

The genome sequencing data generated in this study have been deposited in the SRA database under accession code SRR24464804. Source data are provided with this paper.

## Code availability

All scripts used in image analysis are available at https://github.com/veeninglab/Capsulator and at Zenodo[68].

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

## Acknowledgements

Work in the Veening lab is supported by the Swiss National Science Foundation (SNSF) (project grants 310030_192517, 310030_200792 and 'AntiResist' 51NF40_180541) and ERC consolidator grant 771534-PneumoCaTChER. This work was also funded by NIH grants, R01 AI150893 and R37 AI038446, to J.N.W. We thank Dirk van Swaay (Wunderlichips) for design and production of microfluidics chips. We thank Paddy Gibson for help in genome sequence analysis.

## Author contributions

A.S.R. and J.W.V. wrote the paper with input from all authors. A.S.R., R.V.R., S.D.A., and J.S.M. performed the experiments. A.S.R, R.V.R., S.D.A., G.L., Y.S., J.N.W., and J.W.V designed, analyzed and interpreted the data.

## Competing interests

The authors declare no competing interests.
