## [Peer Review File · Nature Communications]

Synthetic genetic oscillators demonstrate the functional importance of phenotypic variation in pneumococcal-host interactionsREVIEWER COMMENTS

Reviewer #1 (Remarks to the Author):

Review for Rueff et al.: "Rewiring capsule production by CRISPRi-based genetic oscillators demonstrates a functional role of phenotypic variation in pneumococcal-host interactions"

This manuscript introduces a synthetic three-node CRISPRi-based ring oscillator to investigate the significance of phenotypic heterogeneity in pneumococcal capsule production under host-related conditions. The study demonstrates that pneumococcus bacteria with variable capsule levels exhibit an advantage in host colonization, when compared to non-variable mutants, suggesting the relevance of heterogeneity to pathogenic virulence. Overall, this work is very interesting and contributes to the research field of phenotypic heterogeneity in bacteria. Furthermore, as engineering of pneumococcus is challenging, it adds to the type of bacteria in which heterogeneity can be studied, beyond the well-studied model bacteria. It should therefore appeal to a broad audience spanning synthetic biology to microbial physiology.

Minor revisions

- Repeating the experiments in Figure 3 with a construct that does not contain the *cps* operon but another unrelated sequence, a control sequence. This can emphasize that the differences in biofilm formation, adherence to epithelial cells and mice colonization in the different constructs used in Figure 3 are indeed because of differences in capsule production and not because of differences in the construct itself (for example not producing *mScartetI* in the CAPSULATOR-OFF strain).
- In order to emphasize that the oscillator comes from the construct, it would be good to add the autocorrelation plots also of the capsulator-on and off strains.
- It would be important to mention also the growth rate of the various strains in rich medium

Additional points

1. Fonts in Extended Fig. 1a better to have the same font size between matching compounds in the upper and lower panels.
2. Colours in Extended Fig. 1b are hard to distinguish. Better to change them to more contrasting colours.
3. Figure 1c: what are the measurement units of the numbers? hours? I think it does not also appear in the figure legend.
4. Figure 3: please clarify to which statistical significance, i.e., to which p-values the *, **, *** refer to.
5. Lines 319-320: "Over the course of this study, the GRN behaved as expected." What does this mean?
6. Lines 322-323: "It is interesting to note that wild type bacteria outperformed CAPSULATOR bacteria during murine colonization (Fig. 3d)". In Figure 3d it appears that the difference between the wt and the CAPSULATOR is not significant. Please clarify. The authors further write (lines 324-325): "This implies that we have not captured the ideal expression dynamics of *cps* and that precise regulation of capsule synthesis in response to the environmental conditions is crucial for optimal colonization.". In case the change is not significant, is this sentence correct?
7. Fig. 3b: it seems that the Capsulator-ON has lower capsule levels than the oscillating capsulator; why? Isn't it a problem when comparing virulence?
8. Figure 3c: the x-axis is not really a time axis so the figure looks a bit unclear, as if it reflects a time dependence, while the time points are the same for all strains: it would be better to change the x-axis to a non-numerical labelled axis, for example by having bar graphs of different colors grouped into 3 time labels.

Reviewer #2 (Remarks to the Author):

The article by Rueff et al. demonstrates how the use of synthetic biology tools can give insight into phenotypic variation. To show proof-of-principle, the authors designed a synthetic oscillator, characterized by a low metabolic load, in the human pathogen *S. pneumoniae*. The authors

carefully characterized the synthetic oscillator and then coupled its regulatory functions to production of capsule production. In the final sets of the experiments, the authors show that phenotypic variation in capsule production benefits survival and virulence. Overall, the message from this paper is powerful and the results are presented in a clear manner. There is strong evidence that supports the fact that the circuit is stable during multiple cell divisions. However, in order for synthetic circuits to be useful as a broad tool to study phenotypic variation it is important that the authors show and/or reflect on the tunability of the circuit. For example, could what are key design parameters in the CRISPR oscillator that would allow tuning of its frequency. I do not expect that authors show how does influences phenotypic variation, however, I would like to see some experiments along the line of Figure 1 where the authors measure the frequency of the oscillations.

Reviewer #3 (Remarks to the Author):

The manuscript by Rueff and colleagues entitled, "Rewiring capsule production by CRISPRi-based genetic oscillators demonstrates a functional role of phenotypic variation in pneumococcal-host interactions" uses CRISPR technology to dissect the biological function of capsule production variation in *Streptococcus pneumoniae*. The manuscript is exceptionally well-written, clear, easy to follow, and progresses logically. The results provide mechanistic insight into a long-standing question about how capsular variation alters pneumococcal virulence. I have a few comments to help improve the quality of the paper.

The authors label the figures: "Extended Figure" and this is a bit cumbersome in the text. Perhaps just breaking these up into stand-alone figures could provide better clarity.

Extended Fig. 1. Panel b) the Axis labels are somewhat small and hard to read. Perhaps resizing to a larger font might help the reader.

Figure 3, panel D. It might be helpful to include the Δ cps mutant as a control for these experiments and to add more to the discussion about why an acapsular strain has enhanced biofilm and adherence but diminished colonization and survival of starvation.

Figure 3. Why wasn't the CRISPRiLator strain included in the experiments in panels A-C?

A conceptual model would be helpful to integrate the different phenotypes that are governed by capsule variation in this study.

Have the authors taken into account sex as a biological variable with their mice studies? Fetal/neonatal sex in mice can be determined by molecular techniques and this journal is encouraging authors to report data with respect to sex as a biological variable whenever appropriate in vertebrate animal studies.

We would like to thank the referees for their thoughtful and insightful comments and suggestions and were happy to read they appreciated our work. In our opinion, the changes made in response to these points have helped us to develop a clearer and more well-rounded manuscript. We address each comment point by point below.

Reviewer #1:

Review for Rueff et al.: “Rewiring capsule production by CRISPRi-based genetic oscillators demonstrates a functional role of phenotypic variation in pneumococcal-host interactions”

This manuscript introduces a synthetic three-node CRISPRi-based ring oscillator to investigate the significance of phenotypic heterogeneity in pneumococcal capsule production under host-related conditions. The study demonstrates that pneumococcus bacteria with variable capsule levels exhibit an advantage in host colonization, when compared to non-variable mutants, suggesting the relevance of heterogeneity to pathogenic virulence. Overall, this work is very interesting and contributes to the research field of phenotypic heterogeneity in bacteria. Furthermore, as engineering of pneumococcus is challenging, it adds to the type of bacteria in which heterogeneity can be studied, beyond the well-studied model bacteria. It should therefore appeal to a broad audience spanning synthetic biology to microbial physiology.

Minor revisions

- Repeating the experiments in Figure 3 with a construct that does not contain the *cps* operon but another unrelated sequence, a control sequence. This can emphasize that the differences in biofilm formation, adherence to epithelial cells and mice colonization in the different constructs used in Figure 3 are indeed because of differences in capsule production and not because of differences in the construct itself (for example not producing mScarlet1 in the CAPSULATOR-OFF strain).

Indeed, we have performed the mouse colonization experiments also with the CRISPRlator strain as a control. This strain contains the entire three-ring node oscillator as well as the fluorescent reporters. As shown in Figure 3D (now Fig. 5A), this strain shows indistinguishable mouse colonization characteristics compared to wild type *S. pneumoniae* D39V. This control experiment has now been better highlighted in the text for clarity and also based on this we do not think it is necessary to repeat the other experiments (see below and lines 344-347, L464-466, in the tracked changed revised MS).

- In order to emphasize that the oscillator come from the construct, it would be good to add the autocorrelation plots also of the capsulator-on and off strains.

In line with the previous comment, we have analyzed movies from the CRISPRlator control strain and calculated its autocorrelation function. As shown in Figure 1D, the CRISPRlator strain also shows a strong oscillatory pattern demonstrated by the calculated autocorrelation function. Doing this for the ON and OFF strains would be less useful as the mScarlet signal is always ON or always OFF in these cells. In addition, it would encompass generating new long time-lapse experiments for these strains followed by single cell analysis; a large undertaking with a known outcome.

- It would be important to mention also the growth rate of the various strains in rich medium

We have now mentioned the growth rates of the various strains grown in rich C+Y medium in the text (L183-185 in the tracked changed document) and provide the growth curves as a new supplementary figure (Fig. S2e).

Additional points

1. Fonts in Extended Fig. 1a better to have the same font size between matching compounds in the upper and lower panels.

This has now been corrected.

2. Colours in Extended Fig. 1b are hard to distinguish. Better to change them to more contrasting colours.

This has now been corrected.

3. Figure 1c: what are the measurement units of the numbers? hours? I think it does not also appear in the figure legend.

This has now been clarified in the figure and legend. It is indeed hours:min.

4. Figure 3: please clarify to which statistical significance, i.e., to which p-values the *, **, *** refer to.

This has been added to the figures.

5. Lines 319-320: "Over the course of this study, the GRN behaved as expected." What does this mean?

We agree this sentence is a bit vague and this has now been removed. We were very positively surprised that the constructed GRN strains were so stable and well behaved both *in vitro* and *in vivo*. Accumulation of mutations in *dcas9* are very common in other systems. We think that in our system, mutations in *dcas9* are not selected for as that will lead to an ON switch of all the fluorescent reporters simultaneously in our GRNs, which might pose a significant metabolic burden on the cell.

6. Lines 322-323: "It is interesting to note that wild type bacteria outperformed CAPSULATOR bacteria during murine colonization (Fig. 3d)". In Figure 3d it appears that the difference between the wt and the CAPSULATOR is not significant. Please clarify. The authors further write (lines 324-325): "This implies that we have not captured the ideal expression dynamics of cps and that precise regulation of capsule synthesis in response to the environmental conditions is crucial for optimal colonization.". In case the change is not significant, is this sentence correct?

We thank the reviewer for pointing this out as indeed these sentences are not as precise as they should be. Indeed, the difference is not statistically significant. Therefore, we have clarified these statements.

7. Fig. 3b: it seems that the Capsulator-ON has lower capsule levels than the oscillating capsulator; why? Isn't it a problem when comparing virulence?

The reviewer is correct that in this image it seems that capsule production is less in the CAPSULATOR-ON strain. It turned out to be that this image was differently scaled in its signal. This has now been corrected in the revision. When looking at the actual capsule levels as quantified by immunofluorescence, then individual CAPSULATOR-ON cells express similar amounts of capsule as cells that are ON in the CAPSULATOR strain.

8. Figure 3c: the x-axis is not really a time axis so the figure looks a bit unclear, as if it reflects a time dependence, while the time points are the same for all strains: it would be better to change the x-axis to a non-numerical labelled axis, for example by having bar graphs of different colors grouped into 3 time labels.

This is a good point, and we have changed this graph into a bar graph and for clarity only show the 24h time point. This is now Figure 4c in the revised manuscript.

Reviewer #2 (Remarks to the Author):

The article by Rueff et al. demonstrates how the use of synthetic biology tools can give insight into phenotypic variation. To show proof-of-principle, the authors designed a synthetic oscillator, characterized by a low metabolic load, in the human pathogen *S. pneumoniae*. The authors carefully characterized the synthetic oscillator and then coupled its regulatory functions to production of capsule production. In the final sets of the experiments, the authors show that phenotypic variation in capsule production benefits survival and virulence. Overall, the message from this paper is powerful and the results are presented in a clear manner. There is strong evidence that supports the fact that the circuit is stable during multiple cell divisions. However, in order for synthetic circuits to be useful as a broad tool to study phenotypic variation it is important that the authors show and/or reflect on the tunability of the circuit. For example, could what are key design parameters in the CRISPR oscillator that would allow tuning of its frequency. I do not expect that authors show how does influences phenotypic variation, however, I would like to see some experiments along the line of Figure 1 where the authors measure the frequency of the oscillations.

We thank the reviewer for their very positive opinion on our work and it was great to read that the reviewer thinks this is a 'powerful' paper. We do agree with the reviewer that it would be nice to test different versions of the CRISPRi oscillator to find out what the key parameters are that drive oscillation frequency for instance. As suggested by the reviewer, we now reflect on this in more detail in the Discussion of the revised manuscript (L447-454: "It is interesting to note that the frequency of oscillations shown by the single copy, genome integrated pneumococcal CRISPRi and CAPSULATOR GRNs are very similar in length compared to

CRISPRi-based oscillators constructed on multicopy replicating plasmids in *E. coli* (refs 47–49). It is tempting to speculate that dCas9 and sgRNA stability, target binding affinity and DNA replication rates play a role in setting the oscillation frequency. Future research should find out what the key parameters are in driving these CRISPRi-based oscillators so that rational engineering of oscillators of various frequencies and amplitudes would become possible.”). New experimental work to follow up on this will be part of future work and is outside of the scope of the current study.

Reviewer #3 (Remarks to the Author):

The manuscript by Rueff and colleagues entitled, “Rewiring capsule production by CRISPRi-based genetic oscillators demonstrates a functional role of phenotypic variation in pneumococcal-host interactions” uses CRISPR technology to dissect the biological function of capsule production variation in *Streptococcus pneumoniae*. The manuscript is exceptionally well-written, clear, easy to follow, and progresses logically. The results provide mechanistic insight into a long-standing question about how capsular variation alters pneumococcal virulence. I have a few comments to help improve the quality of the paper.

The authors label the figures: “Extended Figure” and this is a bit cumbersome in the text. Perhaps just breaking these up into stand-alone figures could provide better clarity.

We note that this is according to the Nature journal nomenclature, that uses Extended figures to accompany the main figures (instead of Supplementary Figures) to which the paper was originally submitted. However, we thank the reviewer for pointing out that this is not the case for Nature Communications and we agree that sometimes this breaks the flow of the paper, especially the flow between Fig. 1, Extended Figure 2 and Figure 2. Therefore, we now have placed Extended Figs. 2 as regular Figure 2. In addition, we have added a conceptual model in response to Reviewer #3, giving the paper now a total of 5 main figures and supplementary figures (not called Extended Figures anymore).

Extended Fig. 1. Panel b) the Axis labels are somewhat small and hard to read. Perhaps resizing to a larger font might help the reader.

Fixed.

Figure 3, panel D. It might be helpful to include the Δcps mutant as a control for these experiments and to add more to the discussion about why an acapsular strain has enhanced biofilm and adherence but diminished colonization and survival of starvation.

Good point, and in fact we have included a *cps* mutant in this experiment, strain CAPSULATOR- Δcps . This strain is genetically identical to the CAPSULATOR strain beside having a complete deletion of the *cps* operon. We now better discuss why capsule mutants can adhere better but are cleared quicker by the host’s immune system in the revised Discussion (L464-466) and added a clarifying conceptual model (new Fig. 5b).

Figure 3. Why wasn't the CRISPRator strain included in the experiments in panels A-C?

The goal of these experiments was to test which strategy (constitutive capsule production, absence of capsule or heterogeneous capsule production) is the most advantageous for biofilm formation, adherence and survival during starvation. Therefore, the CRISPRator strain is not the correct comparator strain to the CAPSULATOR. The experiment shown in panel D is a little more complex as in this case the GRNs are tested *in vivo* where we expect that both adherence as well as immune evasion might be beneficial. Here we also colonized mice with the CRISPRator strain to test whether the synthetic GRN on its own, without rewiring capsule production, would have any fitness impact.

A conceptual model would be helpful to integrate the different phenotypes that are governed by capsule variation in this study.

We thank the reviewer for this suggestion and have now generated a conceptual model as new Figure 5b in the revised manuscript.

Have the authors taken into account sex as a biological variable with their mice studies? Fetal/neonatal sex in mice can be determined by molecular techniques and this journal is encouraging authors to report data with respect to sex as a biological variable whenever appropriate in vertebrate animal studies.

Our previous work has shown that there is no difference in the colonization levels between males and females in our infant mouse model. As such, infants of both sexes were used for our mouse studies, and this has now been clarified in the Methods section of the revised manuscript.

REVIEWERS' COMMENTS

Reviewer #1 (Remarks to the Author):

I have no further comments

Reviewer #3 (Remarks to the Author):

All concerns have been addressed.

REVIEWERS' COMMENTS

Reviewer #1 (Remarks to the Author):

I have no further comments

Reviewer #3 (Remarks to the Author):

All concerns have been addressed.

We thank the reviewers for taking another look at our revised manuscript and were happy to see that they now fully support publication of our work.